# Mercury Content in Central and Southern Adriatic Sea Sediments in Relation to Seafloor Geochemistry and Sedimentology

**DOI:** 10.3390/molecules24244467

**Published:** 2019-12-05

**Authors:** Elisa Droghini, Anna Annibaldi, Emanuela Prezioso, Mario Tramontana, Emanuela Frapiccini, Rocco De Marco, Silvia Illuminati, Cristina Truzzi, Federico Spagnoli

**Affiliations:** 1Department of Pure and Applied Sciences (DiSPeA), University of Urbino Carlo Bo, Campus Scientifico E. Mattei, 61029 Urbino, Italy; elisadroghini84@gmail.com (E.D.); mario.tramontana@uniurb.it (M.T.); 2Department of Life and Environmental Sciences, Università Politecnica delle Marche, Via Brecce Bianche, 60131 Ancona, Italy; emanuela_prezioso@libero.it (E.P.); s.illuminati@univpm.it (S.I.); c.truzzi@univpm.it (C.T.); 3CoNISMa, Consorzio Nazionale Interuniversitario per le Scienze del Mare, Piazzale Flaminio, 9, 00196 Roma, Italy; 4Institute of Biological Resources and Marine Biotechnology (IRBIM), National Research Council (CNR), Largo Fiera della Pesca, 2, 60125 Ancona, Italy; emanuela.frapiccini@cnr.it (E.F.); rocco.demarco@cnr.it (R.D.M.)

**Keywords:** mercury, sediments, grain size, organic matter, Adriatic sea

## Abstract

Mercury contents were determined in surface sediments from the Central and Southern Adriatic Sea to gain insight into the processes, factors, and variables affecting its distribution. Mercury concentration was measured by thermal decomposition amalgamation atomic absorption spectrometry in samples collected by box-corer from Ancona to Santa Maria di Leuca during the CNR-PERTRE cruise (16/9-4/10/2016). Sediments were also evaluated for chemical-physical parameters (pH, Eh), biogeochemical composition (total carbon, inorganic carbon, total organic carbon, organic matter) and grain size. The average mercury concentration in the Adriatic Sea sediment was 0.053 mg/Kg (d.w.), range 0.011–0.12 mg/Kg (d.w.). Mercury content was mainly affected by grain size and organic matter (OM) distribution, whereas anthropic factors exerted a limited influence. Concentrations followed the distribution of sediment types (clay > silt > sand) due to Adriatic Sea hydrodynamics and were well below the regulatory limits in all samples.

## 1. Introduction

In marine ecosystems, the contaminant load is highest in sediment, which is the ultimate sink for the particulate material circulating in the overlying water column. The contaminants adsorbed to or incorporated in particulate material are transferred by sedimentation to the seafloor, which in turn becomes a potential source of pollution for the overlying aqueous matrix.

Pelagic and benthic aquatic ecosystems are characterised by four closely interacting compartments, the water column, suspended material, sediments, and pore waters. In particular, adsorption/desorption and co-precipitation between suspended particulate and the dissolved phase are strongly influenced by pH, salinity, and oxidation-reduction conditions. Particulate material and bottom deposits are closely interconnected through erosion and sedimentation processes [1,2].

Sediments form a complex, time-varying matrix [3,4], where biogeochemical, biological and bioturbation processes influence the concentration of pollutants. The early centimetres of surface sediment represent the active portion interacting with the water column [5], whereas the deeper layers have no interactions with the bottom interface and provide a record of past events. 

The deposition of marine sediments is influenced by accumulation rate, hydrological and biogeochemical processes, as well as anthropogenic inputs. In semi-enclosed environments, the low-energy, confined circulation may favour contaminant accumulation [6].

The main pollutants found in marine sediment include heavy metals such as mercury (Hg). Mercury distribution is governed by a complex biogeochemical cycle that involves the whole of nature, i.e., the atmosphere, the hydrosphere, and the geosphere, where its accumulation is a recognised health risk worldwide. The toxicity of its organic form, methylmercury (MeHg), its ability to interact with biological components, and the risk of bioaccumulation in the aquatic food chain make it a Water Framework Directive priority substance [7] in which its quantitative analysis in marine sediments is of primary importance.

Mercury is a highly toxic element that has a natural and anthropic origin. It is naturally present in the Earth’s crust (about 0.08 parts per million [8,9,10]). In aquatic environments, it is readily transformed into organomercury compounds such as MeHg by chemical and biological (i.e., bacteria-mediated) pathways that affect its solubility, volatility, bioavailability and toxicity. Methylmercury can be bioaccumulated along the food chain more efficiently than other trace metals; it is the most toxic mercury species and exerts numerous adverse effects, including neurotoxicity, genotoxicity, and endocrine disruption on a wide range of vertebrates, including fish, and invertebrate species [11].

Numerous and disseminated anthropogenic sources—especially mining, gold mining, processing of Sulphur minerals, paint production, woodworking, paper, sod, and chlorine production—account for two-thirds of the mercury found in the terrestrial environment. The remainder is due to pre-industrial flows of natural origin, volcanic activity, and forest fires, which compound the biological activity and gaseous flows from natural waters. The absorption of mercury to soil and marine sediments is influenced by their organic matter (OM) concentration, especially where chlorine species are scarce and hydroxide species predominate [12]. According to previous authors [13], about 85% of the mercury contained in sediments is associated with OM, which is rich in proteins containing -SH groups and humic acids formed following the microbial biodegradation of OM.

The Mediterranean Sea is characterised by large deposits of cinnabar (HgS) that account for about 65% of the world’s mercury reserves, although the Mediterranean represents only about 1% of the world’s oceans. In this area, the sources of mercury contamination are, therefore, both natural and anthropogenic ([14] and references therein). 

Data on mercury concentrations in the Adriatic Sea are limited both as regards water and sediments. Water concentrations have been investigated by Ferrara and Maserti (1992) and Faganeli et al. (2003) [15,16]. Sediments have been investigated by Fabbri et al., 2001; Covelli et al., 2001; Piani et al., 2005; Acquavita et al., 2012 [17,18,19,20,21]. These papers provide evidence that the highest values occur in the Northern part of the Adriatic basin and are mainly related to anthropic activities.

This study was aimed to measure mercury concentrations in Central and Southern Adriatic Sea surface sediments in order to evaluate the natural (sedimentological, geochemical, and biological) and anthropic factors that influence them.

## 2. Study Area

The Adriatic Sea is a semi-enclosed, elongated basin extending in the NW–SE direction for about 800 km between the Italian peninsula to the west and the Balkan Peninsula to the east [22,23,24]. The Northern Adriatic Basin consists of a large continental shelf that represents the shallowest area of the Adriatic Sea [25]. The Central Adriatic is deeper and is characterised by the Middle Adriatic Depression (MAD; slightly >250 m deep), which extends in the NE–SW direction perpendicular to the major axis of the Adriatic basin and to the tectonic and structural orientation of the Apennines ([26] and references therein). The Southern Adriatic is characterised by a deep depression (the South Adriatic Depression; SAD), a sub-circular bathyal basin with a maximum depth of less than 1300 m. Sediment transport in the Adriatic Sea is strongly influenced by wave motion and sea currents, in which the predominant cyclonic circulation favours sediment fluxes with a mainly longitudinal dispersion [27,28] (Figure 1).

Generally, on the Italian side, the coastal sediments are characterised by coarse material that becomes clayey and then sandy as one moves east towards the centre of the basin [22]. The coastal sediments consist mainly of materials carried to the sea by the rivers and reworked by the waves at shallow depths. The clayey sediments form a depositional body (Holocene mud belt) resulting mainly from the deposition of fine materials carried by the Po River and the Apennine rivers [29,30] (Figure 2). 

The minerals identified in sediments reflect the lithological and compositional characteristics of their areas of origin [27,28,31,32]. Two major sediment fluxes can be recognised: (i) the “Apennine flux”, near the coast, begins south of the Po delta and collects clayey-silty sediments carried by the Apennine rivers, which are mainly characterised by a high smectite content; (ii) the “Po River flux” carries sediments transported by the Po River along the basin, which are characterised by clayey-silty sediments, which are rich in illite, chlorite, and kaolinite. 

The sediments found on the Northern and western side of the Adriatic Sea can generally be traced to four main regions that correspond to drainage areas represented by the eastern Alps, the Po basin, and the basins of the Apennine rivers to the north and south of the Gargano Promontory [30]. A minor sediment flux in the Southern Adriatic is associated with the rivers flowing on the eastern (non-Italian) side, especially the Albanian rivers [33].

## 3. Materials and Methods

Sediment samples were collected in the Central and Southern Adriatic Sea, from Ancona to Santa Maria di Leuca, during the PERTRE cruise, which was carried out by IRBIM-CNR of Ancona on board R/V G. Dallaporta 16 September to 4 October 2016. The sampling strategy consisted of transects perpendicular to the Italian coast. A total number of 107 sediment samples were collected (Figure 3) using a box corer (10 × 17 × 25 cm). The sediments were first described in terms of macroscopic characteristics, i.e., colour (by the Munsell chart), grain size, and biological features; pH and redox potential (Eh) were measured with punch-in electrodes. Core surface samples (0–5 cm) were then collected using a plastic spatula, divided into aliquots and placed into polyethylene jars for biogeochemical and sedimentological analyses. 

Samples for biogeochemical, mineralogical, and biochemical determinations were stored at −20 °C and freeze-dried. Samples for grain size and water content analysis were stored at 4 °C and dried at 70 °C.

Total mercury content was determined by thermal decomposition amalgamation atomic absorption spectrometry using a Direct Mercury Analyser (DMA-1, Milestone, BG, Italy). Briefly, the samples are heated in a quartz container using compressed air (purity, 99.998%) as the oxidant gas. The mercury vapours pass through a catalyst and the products of combustion are removed and trapped in a gold amalgamator. After high-temperature (850 °C) desorption, the mercury content is determined by measuring absorption at 253.7 μm. The optimised conditions for drying and decomposition (pyrolysis) using 100 mg of sample were respectively 250 °C for 60 s and 650 °C for 180 s. Mercury was then quantified by the calibration curve technique (Hg stock solution 1000 mg/L from Carlo Erba Reagents S.A.S, 2% HNO_3_). The mean of at least 3 determinations was used for subsequent analyses; the relative standard deviation was consistently <10%. Detection limit was 3 µg kg^−1^.

The accuracy of the analytical procedure was controlled by analysing the certified reference material PACS-2 for trace metals in sediment. Hg concentration was in agreement with the reference values within experimental errors. Mercury mean value (n = 8) was 2.99 ± 0.02 mg/Kg versus certified values of 3.04 ± 0.20 mg/Kg (Δ (%) = −1.7%). The result was in good agreement with the certified value, and the standard deviation was low, proving good repeatability of the method. The blank values for Hg were found to be negligible compared to the lowest concentration to be determined.

The carbon in the sediments of the PERTRE cruise has been determined as total carbon (TC), total inorganic carbon (TIC) that is essentially the carbon linked to the carbonatic fraction of the sediment, and total organic carbon (TOC) is the carbon linked to organic compounds expressed [34]. The carbon analysis has been carried out using a Shimadzu TOC-VCPH analyser equipped with the SSM-5000A specific module for solid samples at the Marche Polytechnic University, Faculty of Agriculture (Ancona, Italy). The analysis is based on the combustion of a sediment sample that releases the carbon contained in the sample as free CO_2_ gas. The combustion is carried out by a catalytically aided combustion oxidation at 900 °C or 200 °C, depending on the analytical procedure. The carrier gas is 99.9% O_2_ at a flux of 500 mL/min that in this case, also functions as an oxidising gas. The total CO_2_ released by the sample combustion is determined by a non-dispersive infrared system (NDIR). The analysis of each sample takes about 5 to 6 min. The analyser detects a peak whose area is proportional to the content of carbon present in the sample. The content is calculated on the basis of a calibration curve done by standard (Acetanilide and Na-carbonate). The total CO_2_ measured is rated to the weight of the inserted samples so that the concentration value expressed as a percentage is determined.

The total carbon is determined by inserting about 200 mg of bulk sediment in the oven at 900 °C so that the total carbon present in the sample is determined (TC). Another aliquot of 100 mg of the same sample is added with 5 mL of concentrated phosphoric acid (pH < 3) and inserted in the second oven at 200 °C so that the concentration of the carbon linked to the inorganic matter is determined (TIC). Finally, the TOC concentration is determined by the difference between the TC and TIC.

The precision of the analyses has been determined as relative standard deviation (RSD) on three replicates. The RSD for the TOC was <0.05 and <0.04 for the TIC. The accuracy has been determined only on the TC with respect to the standards MESS-2 and PACS-2 (National Research Council Canada) on six replicates. The accuracy was <6% for the MESS-2 and <4% for the PACS-2.

Grain size was determined by wet sieving and an X-ray Sedigraph (Micromeritics 5100, Norcross, GA, USA) analysis. After quartering, the samples were dried in an oven at 70–80 °C for about 48 h. Dry sediment samples were pre-treated with a 16% H_2_O_2_ solution to facilitate the separation of grains and remove the organic matter. Then, the separation from the coarser to finer fraction was carried out by washing with water on a sieve with a mesh of 62.5 µm; the two fractions thus obtained were dried in an oven at 60–90 °C for about 48 h. The coarse fraction (gravel and sandy fraction; >62.5 μm) was separated by sieving, using a pile of sieves (Series A.S.T.M.) arranged on each other from top to bottom with a decreasing mesh, which allowed the separation of the different grain size classes according to the scale of Wentworth [35]. The sediment was put into the first highest sieve and submitted to vibration for two cycles of 6 min; the fraction retained by each single sieve was weighed and expressed in % with respect to the total weight of the sample examined. The pelitic fraction (<62.5 µm) was analysed by a Sedigraph (Micromeritics 5000). Initially, after further quartering, a representative 4 g of sediment were taken. The sediment was treated with a solution of distilled water and sodium hexametaphosphate 6‰ (with anti-flocculant action) for about 24 h. Finally, the sample was subjected to ultrasound for 5–7 min, for further disintegration of any floccules and analysed by the X-ray Sedigraph. The sedigraph measured a granulometric range between 0.1 and 300 µm with a best accuracy of 2%. 

OM was determined by measuring weight loss on ignition by the difference-on-ignition (DOI) method [36]. About 2 g of sediment was weighed in a ceramic crucible are placed in a muffle at 500 °C for six hours and weighed again. OM content is determined by the difference between the weight of the sediment sample before and after the combustion. 

For the weighing, a computerised Mettler AT261 electronic microbalance (readability 0.01 mg, repeatability SD = 0.015 mg) was used. Accuracy tests for the balance were obtained by two certified reference “weights” (OIML class E1) of 10 mg (certified mass 0.0100005 g, 2SD = 0.0020 mg) and 100 mg (certified mass 0.0999979 g, 2SD) 0.0020 mg), respectively. 

Relationships between variables were assessed using Pearson’s correlation coefficient r. All statistical tests were performed with the statistical software STATGRAPHICS (STATGRAPHICS Centurion 2018, Statgraphics Technologies Inc., The Plains, VA, USA).

## 4. Results

### 4.1. Biogeochemistry and Sedimentology

#### 4.1.1. Sediment Distribution

Sediment grain-size distribution (Figure 4) highlighted two major domains located north and south of the Gargano Promontory. The former area is characterised by a narrow coastal strip of sandy-silty sediments and by an offshore clayey-silty belt. Towards the coast, the coarser sediments (sandy fraction, 44–80%) are found in a shallow area feeling the effects of strong hydrodynamics. The finer offshore belt is due to sediments from the Po River and the Apennine river system, which are transported southward by the cyclonic Adriatic circulation [27]. In the centre of the Adriatic, sediment distribution is the result of both present hydrodynamics and the last sea level rise. To the north of the MAD (depth <120 m) the predominant type is coarser and poorly sorted sandy [22] to clayey-sandy sediments (sand, 57–78%). The coarser sediments are due to the reworking of bottom sediments during the last sea level rise. The associated fast transgression induced the formation and progressive migration of littoral systems such as lagoons, beaches and fore-beaches, whose sediments were later modified by the present finer hemipelagic sedimentation and macrobenthic bioturbation. Fine-grained sediments consisting of clay deposits (clay, 60–80%) predominate within the MAD and to the south of this depression, with a clay content peaking in the MAD (approximately 88%). 

Sedimentation in the MAD consists of a continuous succession covering the entire time span embracing the different systems tracts of the last sea level change in shallow waters [38].

Between the Tremiti Islands and the Gargano Promontory, all the collected sediment samples belong to the Holocene mud wedge.

To the south of the Gargano Promontory, the sedimentation has more complex characteristics. The coarser sandy sediments (where present) form a very narrow strip along the coast and are much more abundant (sand, 60–70%) at greater depths (>120 m) to the east and south-east of the Promontory. In the southernmost portion of the coarse shelf, sediments are also found in shallower areas, for instance, off Otranto (sand, 83%) (Figure 4).

#### 4.1.2. pH and Eh

In all samples, pH ranged from 7.03 to 7.80, average 7.45 (Figure 5a). North of the Gargano, Promontory values were slightly lower near the coast (minimum 7.03) and tended to rise in the central areas of the basin (maximum 7.79). South of the Promontory, the pH values were low toward the coast (minimum 7.16) and increased close to the centre of the basin (maximum 7.80). 

The Eh values ranged from −243 to 273 mV (Figure 5b). They were more negative near the coast, indicating anoxic conditions, and higher in the central and deeper parts of the basin; the highest values found at the site on the eastern margin of the study area reflected toxic conditions.

#### 4.1.3. Total Organic and Inorganic Carbon

The TOC values ranged from 0.02% to 1.92% (Figure 6a), although they did not exhibit a clear trend. North of the Gargano Promontory values were highest in the MAD and in two areas close the coast (1.55% and 1.77%); they were intermediate in the Holocene mud belt, and lower in the area characterised by relict sands and in two areas closer to the coast (<LOQ and 0.17%). South of the Promontory, TOC values were lower (<LOQ) in sandy sediments and higher (1.81%) in clayey sediments. 

North of the Gargano Promontory, the IC values were higher near the coast (5.76%) and lower towards the central and deeper areas of the basin (3.31%) (Figure 6b). Furthermore, they increased in the areas characterised by relict sands (depth, <120 m) and in the morphological highs in the central basin (Pomo Island and Gallignani sills). South of the Promontory, values were low near the coast (3.57%) and in the shelf area between Bari and Otranto (3.32%) and high in the offshore areas of the Northern sector, which is characterised by coarse sandy deposits (4.90%) (Figure 4). 

#### 4.1.4. Organic Matter

The OM content ranged from 1.49% to 14.51% (average 10.22% *w*/*w*) in all samples (Figure 7). Its areal distribution was similar to that of TOC: north of the Gargano Promontory, its concentrations were lower near the coast (<2%), higher in the MAD (13.99%), and lower (range: 3.08–6.57%) in the shallow areas of relict sands and in the central Adriatic basin. South of the Promontory, concentrations were higher in the shelf area off Trani, Brindisi and Lecce (6.34%) and low (3.36%) on the north-western border of the SAD, between the area off Gargano Promontory and Ostuni.

### 4.2. Mercury

In the collected samples, mercury concentrations ranged from 0.01 to 0.12 mg/kg (average 0.05 mg/kg) (Table 1) and were well below the 0.3 mg/kg limit set by the Environmental Quality Standard Water Framework (Directive 60/2000 [7]). 

The Hg concentration areal distribution (Figure 8) from Ancona to Vieste was characterised by three contiguous belts from the coast to the offshore, which exhibited respectively medium values (coastal area; range 0.01–0.03 mg/Kg), high values (central belt; range, 0.04–0.12 mg/Kg), and low values (easternmost area; range, 0.01–0.02 mg/Kg). In the easternmost area, values were lowest in the sector characterised by relict sands (range, 0.01–0.04 mg/Kg). In the shelf area south of Vieste, mercury concentrations declined from the coast (0.06 mg/Kg) to offshore areas (0.04 mg/Kg). 

The highest concentrations were recorded North of San Benedetto del Tronto (station S23, 0.10 mg/Kg), to the SE of Tortoreto (0.12 mg/Kg), and south of Otranto (0.12 mg/Kg). 

## 5. Discussion

The mercury distribution in marine sediments depends on both the initial chemical form in the original rocks and the transportation and deposition processes controlling the sediment grain size. In particular, similar to other metals, mercury shows high affinity for finer sediments (<63 µm or <20 µm) [42,43,44] and OM [45,46]. Mercury is absorbed into the clay fraction through ion exchange, active sites, and organic substances [47]. 

The areal distribution of sediment types and mercury concentrations is shown in Figure 9, which highlights the strong correlation between finer sediments and higher mercury concentrations; this is also confirmed by Pearson values (*p* = 0.00001). Furthermore, the highest mercury concentrations were measured in the samples with the highest OM content (Figure 9b). According to Crecelius et al. (1975) [13], about 85% of mercury in sediments are associated with OM, due to its high content in proteins containing -SH groups and humic acids.

The strong influence of sediment type on mercury distribution involves a direct role for sedimentary processes, which, in turn, are closely dependent on local hydrodynamics and determine grain size distribution and sediment accumulation rates. In particular, the strong hydrodynamics characterising the coastal areas (e.g., wave motion and rip currents) affects bottom grain size and exerts a strong selection of the sediments carried by rivers, favouring the deposition of coarser-grained materials on the shallower bottoms (e.g., submarine beaches) and the transport of the finer particles to offshore areas. Present and relict sediments were encountered in the offshore areas. The relict sediments of the Northern Adriatic shelf and of the outermost portions of the Apulian continental shelf are similarly coarse-grained deposits lying in areas, which are currently characterised by low-energy hydrodynamics and are related to ancient depositional processes due to the most recent sea level fall and successive sea level transgression. In contrast, present and recent (Holocene) deposits in the deepest areas of the Adriatic basin and in several shelf areas are constituted of pelites reworked and deposited by present currents [29,48,49].

The mercury distribution, in line with the above considerations, is highest in fine Holocene sediments (Figure 9a). Therefore, high mercury concentrations were found in the fine-grained Holocene mud belt offshore Southern Marche and in the inner shelf area of the southernmost Apulian sector. The high concentrations measured in the offshore samples from Marche and north of the Gargano Promontory (Figure 9a) are due to inputs from the north since fine sediments from the Po River and the Apennine rivers are transported south by the current system flowing off the Italian coast. The high mercury contents are likely related to anthropic factors, also considering the strong industrial development that has involved the Po Valley in recent decades. In the inner shelf area of Southern Apulia, which is not influenced by the Northern Adriatic general circulation, higher mercury values are due to local factors, also including a contribution from anthropogenic sources. 

Comparison of the present data with earlier results from sediments with similar grain size and sampling thickness (maximum 4 cm) highlighted comparable and slightly lower mercury concentrations in our samples in the same areas [17,18] [39,40,41] (Table 1). In contrast, our values are much lower than those measured in highly impacted areas such as the Gulf of Trieste and Marano Lagoon [18,19] and Rijeka harbour [50].

The mercury concentrations were highest near busy harbours and highly industrialised areas, which are heavily affected by maritime traffic and other human activities, and sometimes exceeded regulatory limits force (0.3 mg/Kg, [7]). According to Dolenec et al. (1998) [39], most of the mercury in the Central Adriatic derives from the Northern Adriatic area, with a minor influence from the eastern side. Possible sources on the eastern side areas are Kaštela Bay near Split [51,52] and Valona Bay, which are found on the edge of the SAD [53]. In this work, all mercury concentrations were below the Water Framework Directive Environmental Quality Standard (2000) [7].

## 6. Conclusions

In the present study, mercury concentrations were determined in surface sediment collected in the Central and Southern Adriatic Sea to investigate their relationships with grain size and depositional processes.

Mercury concentrations ranged from 0.0106 to 0.123 mg/Kg (average 0.0526 mg/Kg) and were, therefore, well below the 0.3 mg/kg limit set by European regulations (WFD 2000/60 [7]). 

Concentrations were strongly influenced by sediment grain size and organic matter content (Figure 9a,b), they were higher in fine sediments with a relatively high organic matter content and lower in coarser sediments (sand and silty sand). These findings indicate that grain size is the key factor affecting mercury distribution in the Central and Southern Adriatic Sea because a finer grain type involves a larger number of specific absorption sites and favours organic matter accumulation and conservation. Accordingly, concentrations were highest in the areas characterised by pelites of the Holocene mud belt and of the MAD deposits and lower in coarser sediments near the coast and in offshore relict sands. High values found in some shelf areas in the Southern Adriatic may be due to local, probably anthropogenic sources.

In conclusion, Central and Southern Adriatic Sea sediments are not contaminated with mercury, in which low concentrations easily meet the Good Environmental Status objectives of the Marine Strategy Framework Directive (2008/56) [54]. The present study furthers our knowledge of the processes that control mercury concentrations in the sea.

## Figures and Tables

**Figure 1 molecules-24-04467-f001:**
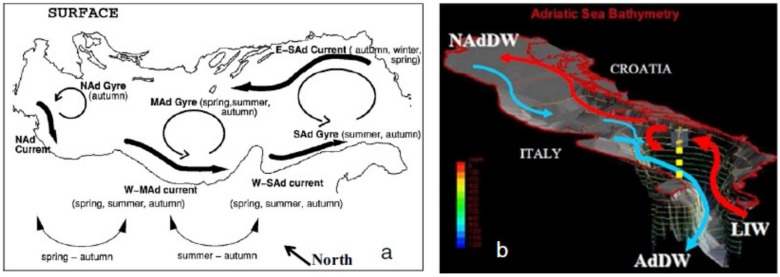
General circulation in the Adriatic Sea. (**a**) Main surface summer currents; (**b**) main winter and spring hydrodynamic. LIW = Levantine Intermediate Water (LIW). AdDW = Adriatic Deep Water. NAdDW = North Adriatic Deep Water [27].

**Figure 2 molecules-24-04467-f002:**
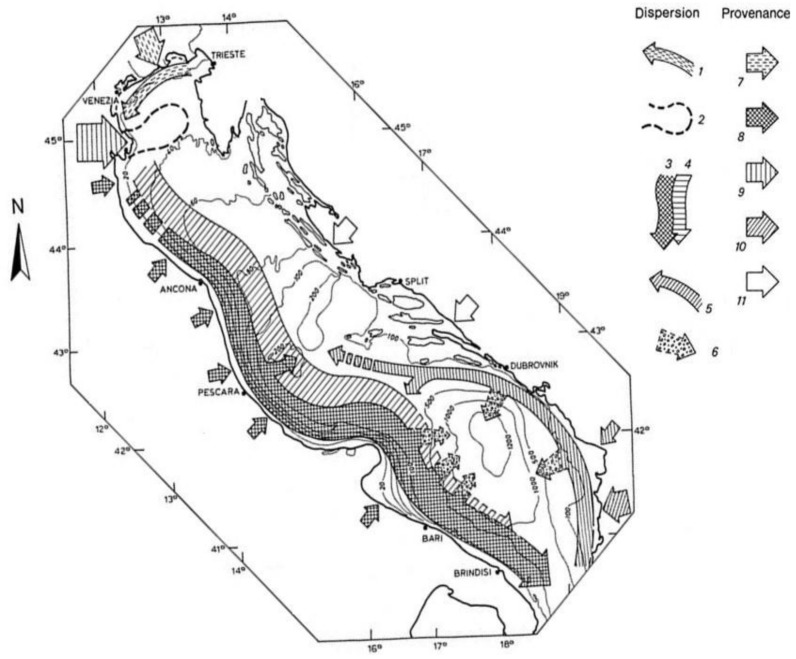
Origin and dispersion pattern of clayey sediments in the Adriatic basin: (1) local dispersion of sediment supply from the rivers of the Veneto province; (2) plume of the River Po; (3) Apennine flux; (4) River Po flux; (5) Albanian flux; (6) turbidite currents; (7) contributions from the rivers of the Veneto province; (8) contribution from the Apennine rivers; (9) contribution from the River Po; (10) contributions from the Albanian rivers; (11) contributions from the Dalmatian rivers [22,27].

**Figure 3 molecules-24-04467-f003:**
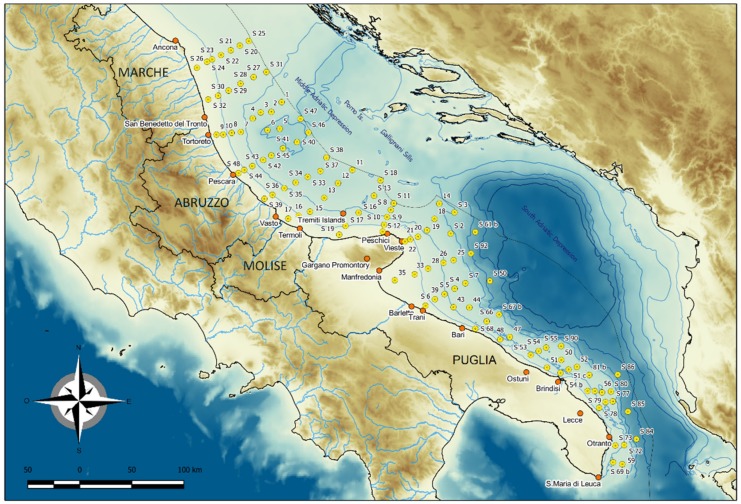
Location of the sampling stations in the PERTRE Cruise. Bathymetric data from GEBCO 2014 [37].

**Figure 4 molecules-24-04467-f004:**
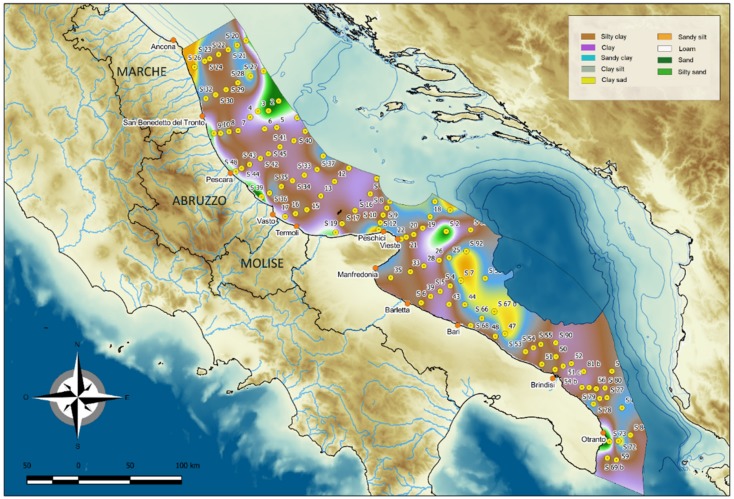
Shepard class size sediment distribution on the western (Italian) side of the Central and Southern Adriatic Sea.

**Figure 5 molecules-24-04467-f005:**
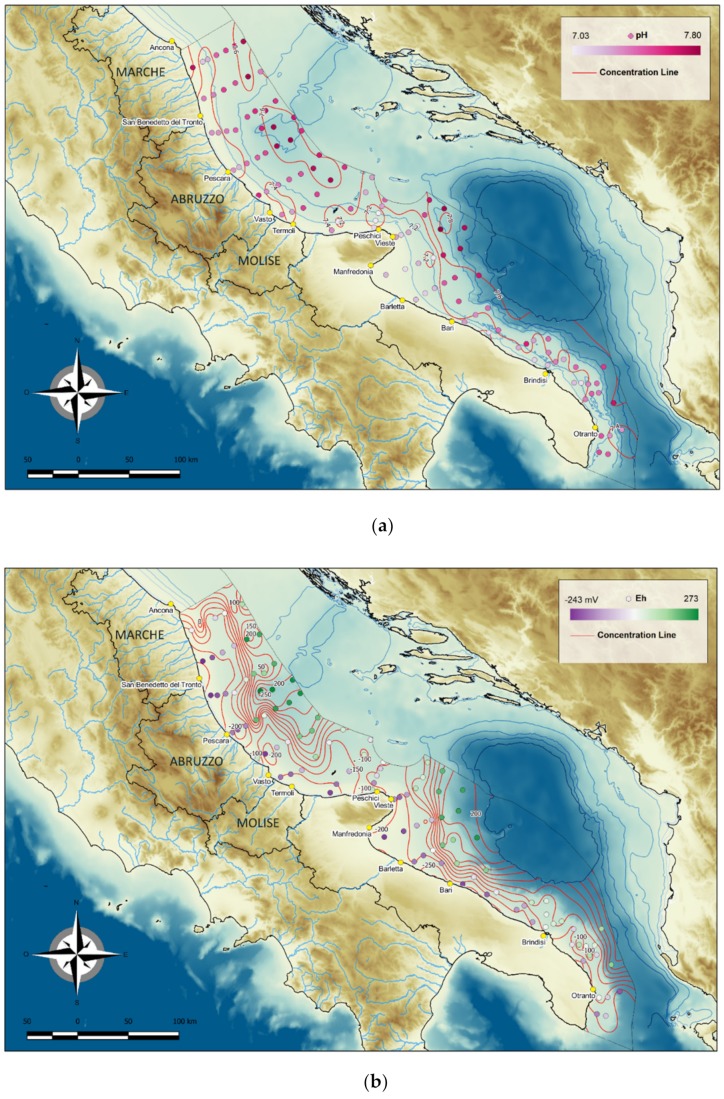
Distribution of pH (**a**) and Eh (**b**) in surface sediments of the Central and Southern Western Adriatic Sea.

**Figure 6 molecules-24-04467-f006:**
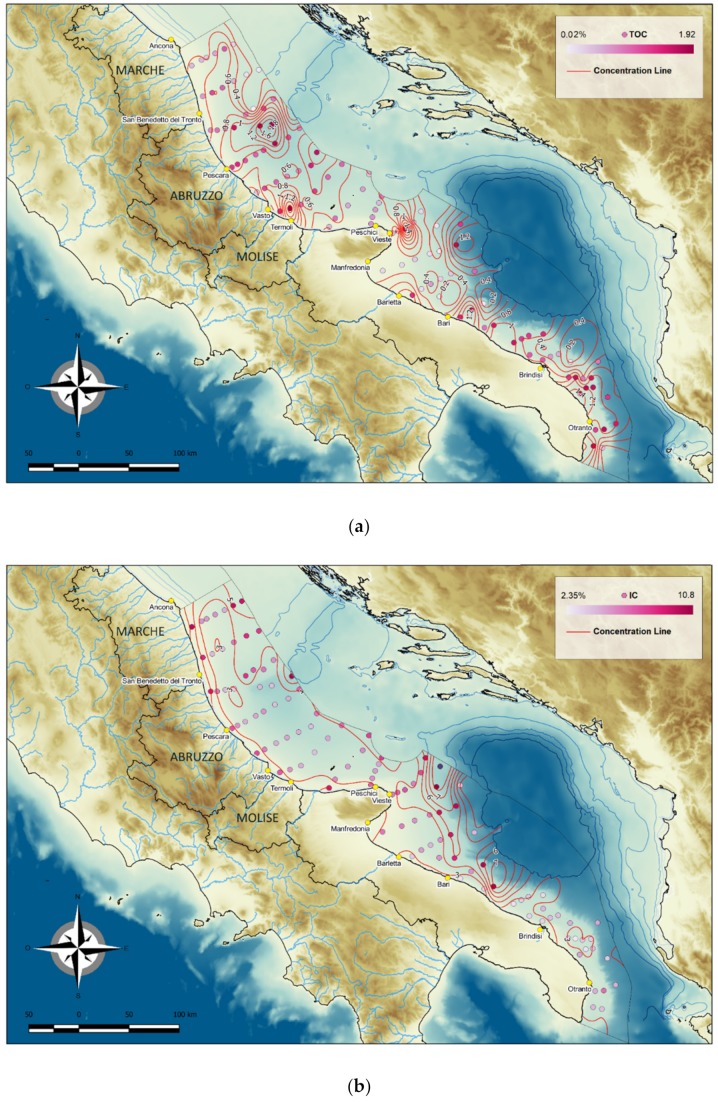
Distribution of total organic carbon (TOC) (**a**) and inorganic carbon (IC) (**b)** in surface sediments of the Central and Southern Western Adriatic Sea.

**Figure 7 molecules-24-04467-f007:**
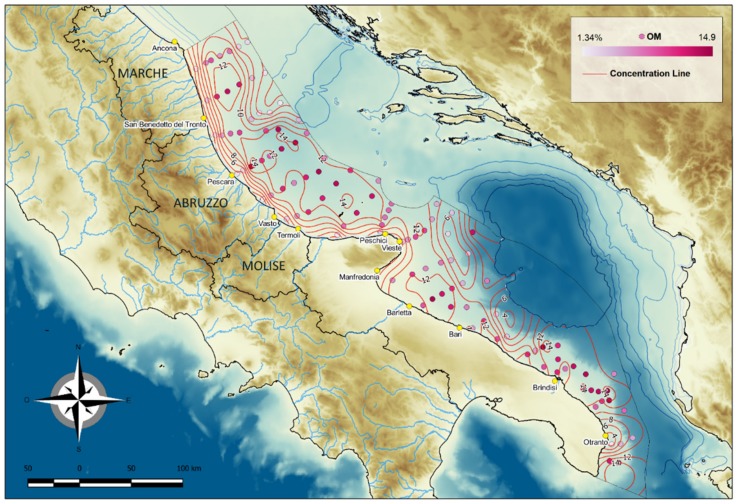
Distribution of OM in surface sediments of the Central and Southern Western Adriatic Sea.

**Figure 8 molecules-24-04467-f008:**
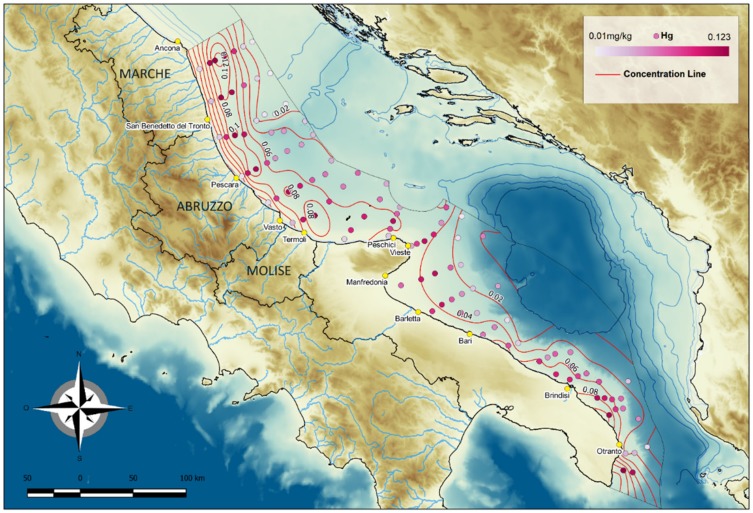
Mercury distribution in surface sediments of the Central and Southern Western Adriatic Sea.

**Figure 9 molecules-24-04467-f009:**
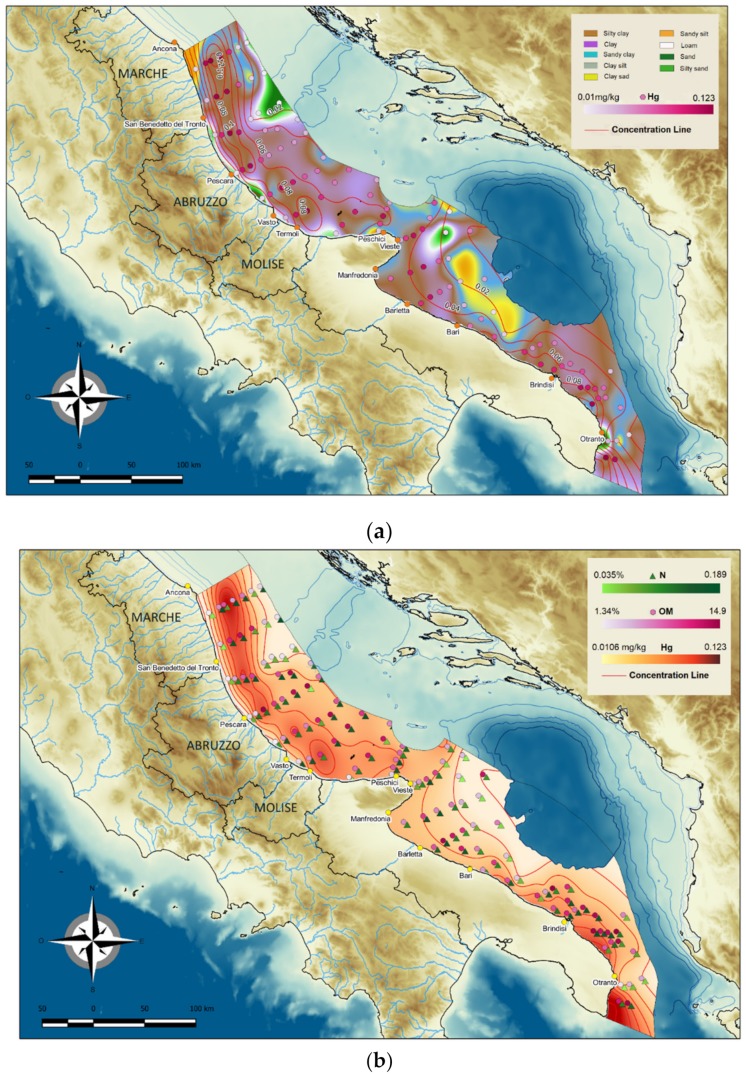
Areal distribution of mercury concentrations in relation to (**a**) sediment type and (**b**) organic matter content.

**Table 1 molecules-24-04467-t001:** Average and range of the mercury concentrations (Hg, mg/Kg d.w.) in the different types of sediment of the Adriatic Sea and comparison with literature data (for detailed data, see Appendix A in Appendix A). EQS (2000)—Environmental Quality Standard from the Water Framework Directive, 2000 [7].

Sampling Area	Gran Size	Hg [mg/Kg d.w.]		Bibliography
		Average	Range (min–max)	
Northern Adriatic Sea	n.a.	0.40	0.09–1.2	[39]
Central Adriatic Sea	n.a.	0.10	0.02–0.25	[39]
Southern Adriatic Sea	n.a.	0.10	0.07–0.42	[39]
Gulf of Trieste	n.a.	5.24	0.10–23.30	[18]
Marano and Grado Lagoon (Northern Adriatic Sea)	n.a.	4.07	0.13–6.58	[19]
Po river (Northern Adriatic Sea)	n.a.	0.05	0.07–0.23	[17]
Rijeka harbour (Croatia)	n.a.	2.50	0.1–8	[40]
Bakar Bay (Croatia)	n.a.	0.45	0.3–0.7	[41]
Rijeka Bay (Croatia)	n.a.	0.03	n.a.	[40]
Central Adriatic Sea	sand	0.02	0.01–0.04	This work
	silty sand	0.02	0.01–0.02	
	clayey sand	0.02	0.022–0.023	
	sandy silt	0.02	0.026–0.021	
	clayey silt	0.05	0.03–0.06	
	clay	0.05	0.04–0.09	
	sandy clay	0.04	n.a.	
	silty clay	0.08	0.06–0.12	
	loam	0.03	0.02–0.04	
Southern Adriatic Sea	sand	0.03	0.02 -0.04	This work
	silty sand	n.a.	/	
	clayey sand	0.03	0.01–0.04	
	sandy silt	n.a.	/	
	clayey silt	0.04	n.a.	
	clay	0.06	0.04–0.08	
	sandy clay	0.05	0.03–0.1	
	silty clay	0.06	0.02–0.12	
	loam	n.a.	/	
Limit value of Hg	/	0.30	/	[7]

n.a., not available.

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
