# Peer review of "Mercury Content in Central and Southern Adriatic Sea Sediments in Relation to Seafloor Geochemistry and Sedimentology"

_molecules, 2019, doi:10.3390/molecules24244467_

Round 1
Reviewer 1 Report
Droghini et al. present an extensive dataset of mercury concentrations in surface sediments collected from Central and Southern Adriatic Sea, and also characterize the sediments in terms of grain size, mineral composition, pH and Eh values, and carbon contents including TC, IC, TOC, and OM concentrations. They found that mercury contents are mainly controlled by grain size and OM distribution, consistent with previous studies in the region although the values are lower. The mercury pollutants absorbed in the sediments are much lower than the regulated value (0.3 mg/kg), indicating that the sediments were not contaminated by human activities. However, an interesting question arises that the average Hg content (0.05 mg/kg, or ppm) is also lower than the crust Hg abundance (~0.08), suggesting that mercury is actually depleted in the sediments. This seems abnormal when considering that the north of this region accounts for 65% Hg reserve in the Earth.
Albeit the question, this study is well organized and written, providing valuable data or information that would be important to understand mercury circling in the ecosystem, in particularly in the Adriatic Sea that appears to be able to self-clean up this toxic metal via natural processes.
A number of minors are picked up and marked in the PDF manuscript for consideration of the authors if a revision is made.

Author Response
As regard the average Hg content (0.05 ppm) that is lower than the mean crust abundance (0.08 ppm) It should be said that the values of 0.08 ppm refers to an average that includes different type of rocks comprising also the magmatic rocks that naturally present higher values of Hg while the sediments are dealing with our paper refers mainly to sedimentary rocks that naturally present depleted values of Hg. To support this can be cited: the work of Ferrara and Maserti (1991) that present range of values concentration if surface sediments of 0.02-0.13 for Central Adriatic and of 0.03-0.07 ppm for Southern Adriatic; the paper of Fowler et al (2000) that report ranges of 0.00086 and 0.004 ppm; Frontalini and Coccioni (2007) that report mean value of 0.03 for the coastal Central Adriatic Sea; and also other authors that are in line with our values.
Besides the 65% of the mercury mentioned in the text refers to the whole Mediterranean area and not to the only Adriatic Sea. Therefore, our results are consistent with these references.
Minor revisions are corrected in the text and in the attached PDF.

Reviewer 2 Report
Manuscript "Mercury Content in Adriatic Sea Sediments in Relation to Seafloor Geochemistry and Sedimentology “ authors: Elisa Droghini, Anna Annibaldi, Emanuela Prezioso, Mario Tramontana, Emanuela Frapiccini, Rocco De Marco, Silvia Illuminati, Cristina Truzzi, Federico Spagnoli; present interesting research results and is very well written.
In title I suggest that authors include: central and southern Adriatic instead of Adriatic.
Authors mention different geographic locations (Santa Maria di Leuce, Tremiti Islands Gargano Promonrory, Trani, Lecce, Ostuni, Potenza Picena, Tortoreto, Apulia)) that are not visible in any of the images. Insert locations in Figures or explain in text.
Line 89 - …depth is about 1300 m – please replace about with less than
Line 110 – (6) Turbine current?? or Turbidite current
Line 222 - … poorly sorted sandy to clayey – sand sediments – How authors know that is poorly sorted sediment? Insert reference
Line 226-228 – sentence: Fine-grained…..in the MAD and to the south, the clay content peaking… in the MAD. Please rewrite
Line 322 – the sentence mentions x three times - check
Line 399-401 - sentence: Comparison… Insert reference
Author Response
We changed the title as suggested.
We added the geographic locations mentioned in the text on the figure 3 and we changed the text deleting Potenza Picena.
Line 89, 110, 222, 226-228, 322 and 399-401 were corrected according to the referee observations.
Reviewer 3 Report
This MS is dealing with the mercury distribution in the surface sediments (regarding granulometry and OM), specifically in the middle and southern Adriatic coastal areas of Italy. Although the subject is important and should be elaborated in the form of the review, I found some parts of discussion poor, which should be improved (see in attached file). While most of the consulted references were used and compared fairly, some other (also in the attached file) should be discussed more critically and in more detail. Also, some corrections are needed for English language and style.
If authors adequately answer all questions raised (here and in attached file), revised MS can be published.
Specific comments are given in the attached file.

Author Response
All questions raised by this referee are point to point in the text and in the attached PDF.
